# Aging and Adiposity—Focus on Biological Females at Midlife and Beyond

**DOI:** 10.3390/ijms25052972

**Published:** 2024-03-04

**Authors:** Amna Rehman, Sanam Lathief, Nipith Charoenngam, Lubna Pal

**Affiliations:** 1Department of Internal Medicine, Berkshire Medical Center, Pittsfield, MA 02101, USA; amnarhm03@gmail.com; 2Division of Endocrinology, Mount Auburn Hospital, Cambridge, MA 02138, USA; sanam.lathief@mah.harvard.edu; 3Department of Medicine, Harvard Medical School, Boston, MA 02115, USA; nipith.charoenngam@mah.harvard.edu; 4Department of Medicine, Mount Auburn Hospital, Cambridge, MA 02138, USA; 5Department of Medicine, Faculty of Medicine Siriraj Hospital, Mahidol University, Bangkok 10700, Thailand; 6Division of Reproductive Endocrinology & Infertility, Department of Obstetrics, Gynecology & Reproductive Sciences, Yale School of Medicine, New Haven, CT 06510, USA

**Keywords:** menopause, estrogens, adiposity

## Abstract

Menopause is a physiological phase of life of aging women, and more than 1 billion women worldwide will be in menopause by 2025. The processes of global senescence parallel stages of reproductive aging and occur alongside aging-related changes in the body. Alterations in the endocrine pathways accompany and often predate the physiologic changes of aging, and interactions of these processes are increasingly being recognized as contributory to the progression of senescence. Our goal for this review is to examine, in aging women, the complex interplay between the endocrinology of menopause transition and post-menopause, and the metabolic transition, the hallmark being an increasing tendency towards central adiposity that begins in tandem with reproductive aging and is often exacerbated post menopause. For the purpose of this review, our choice of the terms ‘female’ and ‘woman’ refer to genetic females.

## 1. Introduction

Reproductive aging is a continuum in either sex. However, the stages of reproductive aging are both overt and well characterized only in genetic females [1,2]. Estrogens (E) are the prototype female hormones. Premenopausal years represent an E replete state. The final menstrual period is a seminal event that, in the absence of intervention, marks the beginning of a phase of life that is characterized not only by an E deplete milieu, but also represents a state of relative androgen excess. Menopause is a physiological phase of life of aging women. The processes of global senescence march in tandem with reproductive aging such that a differentiation between the effects aging from those of menopause on age related changes in bodily function and phenotype is often difficult to appreciate. Over 1 billion women worldwide are expected to be in menopause by 2025 [3]. The prevalence of obesity in older adults continues to increase over time, with women being disproportionately over-represented in populations with obesity. Evidence from recent years suggests that the prevalence of obesity among adults over the age of 60 was more than 37.5% in males and 39.4% in females [4]. A shift in the reproductive hormones during the menopause transition and postmenopause is suggested to contribute to an accelerated rate of weight gain; this phenomenon is particularly prominent during the menopausal transition and in the first few years following the final menstrual period (Figure 1).

Metabolic aberrations such as dyslipidemia, hyperinsulinemia and dysglycemia are intimately intertwined with adiposity and, collectively, these metabolic processes are recognized to both initiate as well as perpetuate common chronic disorders such as cardiovascular disease (CVD) and type 2 diabetes. In this review, we have aimed to examine the available literature on the relationship between chronological and reproductive aging with adiposity in biological women. We review evidence supportive of a regulatory role of E’s, and specifically of 17β-estradiol (17β-E2), the predominant E of premenopausal years, in metabolism and in the female pattern distribution of adipose tissue [5].

## 2. Adiposity

The colloquial term ‘adiposity’ refers to a quantitative excess of adipose tissue in the body [6] and is an eventual outcome when, over time, the caloric intake is disproportionately in excess to the energy expended. The causes are multifactorial: genetic, epigenetic, behavioral and environmental factors have all been shown to be contribute. High adiposity is often attributed to three factors: a high calorie intake, a low basal metabolic rate and a low level of spontaneous physical activity. It is important to note that adiposity is often used synonymously with ‘fatness’ or ‘obesity’. In this review, we restrict the use of the term obesity to elevated body mass index (BMI, in kg/m^2^), a quantifiable, yet fallible, measure that is almost ubiquitously utilized in clinical practice to reflect adiposity [7]. 

Adipose tissue (AT), commonly known as fat, is a type of loose connective tissue. Alongside its primary cell type, the adipocyte, the adipose tissue contains a plethora of additional dwellers, including fibroblasts, vascular endothelial cells, preadipocytes and various immune cells, such as AT macrophages. All the distinct cell populations in the AT are engaged in complex crosstalk pathways [8], functioning to collectively serve as an energy-storage organ. The AT additionally serves as a buffering cushion for bodily organs s well as plays a critical role in thermoregulation [9]. Moreover, in recent years, AT’s role as a distinct endocrine organ has been increasingly recognized [10].

## 3. Adipose Tissue—A Heterogeneous Organ 

Based on the anatomic location, the AT is categorized into subcutaneous and visceral compartments [11]. Subcutaneous adipose tissue (SAT) refers to the fat deposits that exist directly underneath the skin. This is also the compartment where excess *potential energy* is stored. Visceral adipose tissue (VAT) is distinct from the SAT both in anatomical distribution and in function. The VAT is present within the bodily compartments, including intrathoracic, intraabdominal, and pelvic cavities, and is responsive to multiple endocrine and neural signals. It is the VAT that is primarily responsible for obesity-related metabolic derangements and deemed a risk for chronic health disorders including type 2 diabetes mellitus and CVD; excess VAT has additionally been associated with mood and cognitive disorders [3].

The AT is further distinguished into two phenotypically and functionally distinct types: the white adipose tissue (WAT) and the brown adipose tissue (BAT) [12]. The adipocytes in the WAT are unilocular, each characterized by an eccentric nucleus displaced to the cell’s periphery by the large intracellular fat droplets (Table 1). The WAT serves as the reservoir for energy storage in the body and is a source of multiple lipid and protein factors involved in bodily function regulation. During high energy supply/decreased energy expenditure situations, surplus energy is stored in the WAT cells through glucose uptake, lipogenesis and synthesis of triacylglycerols. Conversely, during periods of low energy supply/increased energy expenditure, the WAT cells serve to sustain bodily functions through the release of fatty acids via lipolysis. In an experimental study, individuals with high body fat demonstrated a more pronounced increase in the levels of two adipokines, leptin and resistin, in response to intense exercise compared with those with lesser body fat, suggesting that the adipokine response to exercise is influenced by the AT [13]. In contrast to the white adipocytes of the WAT, the brown adipocytes of the BAT contain smaller lipid droplets and numerous mitochondria, giving BAT its brown color. The brown adipocytes serve as the “bodily furnace”, having a specific physiologic role of utilizing stored energy to generate heat, a process that is referred to as *non-shivering thermogenesis* [14]. This is due to the presence of an increased level of uncoupling protein-1 (UCP-1) in the mitochondria of the brown adipocytes. The UCP-1 catalyzes the coupling of the respiratory chain to ATP synthetase, which facilitates heat production. The BAT is commonly located in specific body regions including the neck, interscapular, supraclavicular and suprarenal areas [15]. It is especially abundant in the newborns, increases into the adolescence, and has been detected on PET scans in adults at “hot spots” in the neck, intercostal spaces near the spine and roots of the upper extremities. In adult humans, the main driver of BAT thermogenesis is increased sympathetic activity, as occurs with noradrenaline release in response to cold. Catecholamine and thyroid hormones, are also implicated in BAT activation; the enzyme *deiodinase* is responsible for the local conversion of circulating thyroxine to triiodothyronine (T3) within the brown and beige adipocytes [12,16,17].

Beige or brite adipocytes are a subclass of WAT cells that are understood to differentiate from a white adipocytic precursor or transdifferentiate from existing mature white adipocytes [12]. Under certain physiologic conditions, such as cold exposure or sympathetic stimulation, the adipocytes within the WAT develop a multilocular appearance and increase their expression of UCP-1 (acquiring features of BAT adipocytes). By doing this, the normally energy-storing adipocytes transform into energy-releasing adipocytes. This process of development of brown-like adipocytes is called ‘browning’ [12,15]. The browning phenomenon is a positive acquired transition in the WAT that can help improve metabolic health. 

Beyond its role in thermogenesis, the AT is a dynamic organ that, through release of a variety of modulatory signals (pro- and anti-inflammatory and endocrine), can influence the overall physiology, as well as has the potential to mediate pathophysiology. The *adipokines* are a variety of peptide and lipid factions that are produced within the AT and are then released systemically and can have autocrine, paracrine and endocrine effects [10,18]. The adipokines consist of the *true* adipokines, which are primarily generated by both pre-adipocytes and mature adipocytes, and the *classical* cytokines, which are produced by the adipocytes as well as the immune cells within the AT depots [18]. Adipocytes express and secrete endocrine hormones such as adiponectin and leptin; however, numerous proteins are also derived from the non-adipocyte fraction of the AT. These two compartments (adipocytes and the non-adipocyte) function cohesively as an integrated unit. Broadly, AT affects the rest of the body through (1) secreted proteins with metabolic activities on remote tissues and (2) enzymes that modulate the steroidogenic pathways [18]. The net balance of the pro- and the anti-inflammatory mediators that are contributed by the WAT is influenced by several factors including calorie intake, metabolic status, oxidative stress, infection, smoking, sex and age [19].

## 4. Quantifying Adiposity in Clinical Practice

Adiposity can be assessed using various anthropometric measures. Table 2 summarizes the common anthropometric measurements utilized in clinical practice as well as common radiological methods that are often employed in clinical research for the quantification of adiposity, with respective strengths and limitations [20]. Body mass index (BMI) is the most common anthropometric measure. It is used as a cut-off marker for weight classification. There are variations in BMI among populations of the same age, gender and level of body fatness [21,22,23,24,25]. The cut-off points for BMI that are specified by the World Health Organization (WHO) are applicable for Caucasian populations but tend to underestimate obesity risk in the Asian and South Asian populations [26]. The WHO classification for BMI and obesity for Caucasian and Asian populations is presented in Table 3 [27,28]. The waist circumference (WC) and the waist-to-hip ratio (WHR) are considered to better correlate with central (visceral) adiposity compared to the BMI. Racial and ethnic differences in WC and WHR are appreciated, and these may be relevant to the recognized racial and ethnic differences in the prevalence of type 2 diabetes and CVD. Compared to Europeans, Asian populations have greater VAT; conversely, African populations and, possibly, Pacific Islanders have less VAT or percentage of body fat at any given WC [29].

## 5. Adipokines and Cytokines—Effectors of Adiposity Related Harm

As previously mentioned, WAT is the source of adipokines [19], which play a major role in the regulation of various physiologic and metabolic processes. In this section, we review the physiology and functions of adipokines and cytokines with respect to energy metabolism. 

**Leptin** (from the Greek ‘leptos’, meaning ‘thin’) is a protein hormone produced by the WAT [30,31]. The most significant roles of leptin include regulation of food intake, maintenance of energy balance and expenditure and improving insulin sensitivity [32]. It serves as a metabolic signal of “energy sufficiency”. Leptin levels drop after weight loss [18]. Leptin also acts on the peripheral adipocytes to stimulate lipolysis. Leptin exerts its biological actions by binding to the leptin receptor (LEPR or OBR). Leptin receptors are expressed in the central nervous system and the periphery [18]. Stimulation of the LEPR receptor causes activation of the JAK-STAT signaling pathway, which induces such actions as an increase in energy expenditure and control of food intake [33]. Other functions of leptin include reduction of stress-induced hypothalamic CRH (corticotropin releasing hormone) secretion and cortisol secretion, stimulation of thyrotropin releasing hormone (TRH) expression and secretion, normalization of suppressed thyroid hormone levels, acceleration of puberty, regulation of hematopoiesis, angiogenesis, bone development and immune function [18]. Leptin replacement during fasting inhibited alteration in the thyroid and hypothalamic pituitary gonadal axes in response to starvation in one study in healthy men [34]. Adipocytes secrete leptin in direct proportion to the AT mass; the secretion is greater from the SAT relative to the VAT [18]. Leptin secretion is increased by estrogens, insulin, and glucocorticoids and decreased by androgens, growth hormone and free fatty acids [18]. Leptin levels are positively related to the amount of AT in the body. In the presence of excess adiposity, leptin gene expression increases in humans [18,35,36]. These alterations lead to hyperleptinemia and leptin resistance. This can be understood by comparing the relationship between elevated plasma insulin levels and the insulin resistance that is observed in the state of obesity. Similarly, in individuals with obesity, weight loss by dieting results in a decrease in plasma leptin [31,37]. One study noted that the total body fat and abdominal obesity, including intra-abdominal AT, was highly correlated to leptin levels in women aged 18–69 years [38].

**Adiponectin** is a cytokine produced in the WAT and other organs including the brain [39]. Adiponectin enhances insulin sensitivity, increases fatty acid oxidation and stimulates glucose utilization [39]. Adiponectin expression is higher in the SAT than the VAT [18]. Adiponectin has unique antidiabetic, anti-inflammatory and antiatherogenic properties [40]. Adiponectin acts mainly via two receptor AdipoR1 and AdipoR2 [41]. The AdipoR1 is present in the skeletal muscle and the AT and acts mainly through an increase in adenosine monophosphate kinase (AMPK) activity. The AdipoR2 is present in the liver and the AT and activates the peroxisome proliferator-activated receptor (PPAR)-α, resulting in improved insulin sensitivity, reduces hepatic glucose output and increases fatty acid oxidation. The blood concentration of adiponectin is inversely correlated to insulin resistance, obesity, and type 2 diabetes [18,42,43,44,45]. Adiponectin levels are suppressed in the presence of insulin resistance due to lipodystrophy or obesity. When insulin sensitivity improves after weight reduction, adiponectin levels increase [18]. The antiatherogenic activities of adiponectin include inhibition of monocyte adhesion, macrophage transformation to foam cells and proliferation of migrating smooth muscle cells in response to growth factors. Furthermore, it increases the production of nitric oxide by the vascular endothelial cells and stimulates angiogenesis [46,47].

**Interleukin 6 (IL-6)** is an inflammatory cytokine that is secreted by several types of cells, such as adipocytes, endothelial cells, immune cells and the myocytes. IL-6 levels in the blood are positively related to BMI, with high IL-6 levels found in obese individuals. The expression and secretion of IL-6 are two to three times greater in the VAT relative to the SAT. There is a positive correlation between IL-6 levels, insulin resistance and obesity and cardiovascular disease [18]. IL-6 receptor and leptin receptors are homologous. IL-6 is involved in multiple physiological pathways, including cell differentiation, inflammation and the tissue healing process. IL-6 acts on AT and stimulates lipolysis, resulting in increased free fatty acid production [48]. It also inhibits adipogenesis and decreases adiponectin secretion [18]. IL-6 also has a central role in that it acts on the hypothalamus to inhibit the appetite [49]. These findings indicate that IL-6 exhibits dual roles in body weight regulation by suppressing appetite in the central nervous system and causing fat loss in the AT [18,50,51]. 

**Tumor Necrosis Factor-α** (**TNF-α**) is a proinflammatory cytokine that is synthesized in the adipocytes. TNF expression is regulated by the total body mass and is greater in the SAT compared to the VAT. Circulating levels and adipocyte expression of TNF-α are increased in obesity and in individuals with metabolic syndrome and positively correlate with insulin resistance [18,51,52]. TNF-α inhibits adipogenesis, stimulates lipolysis and decreases the expression of glucose transporter (GLUT)-4. TNF-α receptors type 1 (TNFR1) and type 2 (TNFR2) are expressed in the AT. Binding to the receptors, TNF-α activates c Jun N-terminal kinase (JNK), IkB kinase (IKK), and mitogen-activated protein kinases (MAPKs). TNF-α also impairs insulin signaling through direct and indirect mechanisms [18].

**Monocyte chemoattractant protein-1 (MCP-1)** is an adipokine that is expressed and secreted by the AT. MCP-1 attracts monocytes to sites of inflammation. Animal studies have shown increased AT expression of MCP-1 and circulating MCP-1 in a rodent model of obesity [51]. MCP-1 also inhibits adipocyte growth and differentiation. MCP-1 directly correlates with AT insulin resistance [18]. 

**Plasminogen activator inhibitor-1 (PAI-1)** is a member of the serine protease inhibitor family and is the primary inhibitor of fibrinolysis [53,54]. PAI-1 is expressed by adipocytes, and its secretion is greater in the VAT relative to the SAT [53]. Plasma PAI-1 levels are positively correlated with obesity, insulin resistance and metabolic syndrome, and predict future risk for type 2 diabetes and CVD [18].

**Resistin** is a polypeptide produced by the AT in rodents and by the macrophages in humans [55]. Animal studies suggest that resistin has significant activities on insulin action, potentially linking obesity with insulin resistance [56]. The relationship between serum resistin levels with insulin resistance, diabetes mellitus, and obesity in humans, however, remains controversial.

**Inflammation**—the eventual mediator of obesity-related harm.

A state of systemic inflammation is a hallmark of excess adiposity. Obesity is characterized by a relative excess of proinflammatory and decreased anti-inflammatory adipokines. Many of the disorders associated with, and attributed to obesity are downstream to insulin resistance and inflammation including endothelial dysfunction and disturbances in glucose homeostasis. The health benefits of fasting and calorie restriction are also attributed to modulations in these signals; in a calorie-restricted state, the anti-inflammatory adipokines are known to rise and the proinflammatory adipokines decline, leading to improvements in insulin signaling immune function [19].

## 6. Sexual Dimorphism and Adipose Tissue

Sexual dimorphism in AT distribution is well recognized. The volume of AT as well as the regional distribution of AT vary between genetic females and males, and the fact that these differences become manifest following attainment of sexual maturity underlines a modulatory role of sex hormones in influencing AT physiology. Reproductive-age women have higher fat mass than age-comparable men [57]. Dr. John Vague was the first to bring to attention a sex differential in the distribution of peripheral body fat [58], and multiple studies have since confirmed this observation across a wide BMI range [57,59,60]. Overall, women have more SAT compared to age-comparable men and, conversely, men have more VAT compared to women. Furthermore, in women, a differential in AT distribution is also apparent by reproductive stage with higher VAT burden in the peri- and postmenopausal compared to premenopausal women. This lesser VAT burden in pre- menopausal women compared to age-comparable men translates to lesser risk of metabolic derangements in premenopausal women compared to men. A role for sex hormones in general and E in particular is recognized as being pivotal to the differential distribution of body fat in the gluteo-femoral (gynoid) region in reproductive-age women (Figure 2). Conversely, a role for androgens is deemed relevant for the android tendency towards central fat distribution, as is common in biological males [57]. Key areas in the hypothalamus, which integrate peripheral adiposity signals, such as leptin and insulin hormones, that are involved in food intake and body fat distribution, are also influenced by sex steroid hormones [61,62].

## 7. Female Reproductive Hormones and Adipose Tissue 

The presence of estrogen receptors (ERα and ERβ) in high abundance in the VAT [63,64,65,66] supports a direct role of E in modulating the function of the visceral adipose tissue. Estrogen is suggested to inhibit lipid deposition within the adipocytes, utilizing two distinct pathways; by directly decreasing lipogenesis through attenuating the activity of the enzyme lipoprotein lipase (LPL), and indirectly by affecting lipolysis through enhancing the function of the lipolytic enzyme hormone-sensitive lipase (HSL) [67,68]. Estrogen also indirectly regulates leptin secretion by the AT as well as influences food consumption and energy expenditure [67]. In addition, in human adipocytes, E_2_ can upregulate α2A-adrenergic receptors (anti-lipolytic) in the femoral and subcutaneous AT, favoring fat deposition in these areas without having any effects on the VAT maintaining high lipolytic activity [68]. These findings help understand the influence of E_2_ in maintaining the female fat distribution. The E_2_ influences the fat distribution by altering the lipolytic response into the two fat depots differently, favoring fat accumulation in the subcutaneous depot at the expense of the visceral depots [68]. Figure 3 provides a schematic summation of the many processes through which estrogens influence the metabolic environment as well as the metabolic consequences of estrogen deprivation. 

## 8. Endocrinology of Reproductive Aging and Adiposity

The actions of estrogen in the AT, the pancreatic beta cells, skeletal muscle, macrophages, liver and the central nervous synergize to enhance glucose and lipid metabolism. Deficiencies or resistance to estrogen in these tissues play a role in metabolic dysfunction, and predisposition to disorders including type 2 diabetes, metabolic syndrome and obesity [67]. A propensity towards weight and fat gain is evident in the perimenopause with evidence of subtle yet progressive changes in female body fat composition towards increasing adiposity as women transition from the pre into the peri and then the menopausal stages of reproductive aging (Figure 4) [69]. Indeed, the period of menopause transition has been associated with an increase in fat mass and body fat percentage, redistribution of fat to the abdominal area, and a decrease in the lean mass. Studies have shown that postmenopausal women have significantly higher VAT than BMI-matched premenopausal women [70,71,72]. On an average, the proportion of VAT in relation to total body fat increases from 5–8% in the premenopause to 15–20% in the postmenopausal stage of life [73].

During the menopause transition, the most notable changes in the reproductive endocrine milieu include a progressive decline in circulating levels of inhibin B and anti-Mullerian hormone (AMH), two hormones originating from the ovarian granulosa cells that belong to the transforming growth factor-β (TGF-β) superfamily of cytokines [74,75]. A decrease in inhibin B levels causes a rebound increase in circulating levels of FSH (follicle stimulating hormone), which is known to be essential for the maintenance of circulating estradiol (E_2_) levels until late in reproductive life when hypoestrogenemia eventually results from the exhaustion of the ovarian follicular reserve [75]. In the post-menopausal state, FSH levels become markedly elevated and E_2_ levels decrease while inhibin B and AMH become undetectable [76]. Despite a marked attenuation in ovarian production of E_2_, the post-menopausal ovaries continue to produce androgens even up to 10 years after the final menstrual period [77]. Androgen production from the ovaries is dependent on ovarian theca cell stimulation governed by the pituitary gonadotropin LH (luteinizing hormone) that rises concomitant with FSH post menopause. It is important to note that reduced E_2_ levels during menopause play a crucial role in the downregulation of the hepatic production of sex hormone binding globulin (SHBG); a decline in SHBG levels follows the cessation of ovarian function in postmenopausal individuals [78,79]. This leads to an increase in the free androgen index (total testosterone/SHBG × 100%) making postmenopause a state of absolute estrogen deficiency and of relative androgen excess. 

A state of relative androgen excess is suggested to be relevant to the postmenopausal shift in regional pattern of fat distribution to an androgenic phenotype [79,80,81]. AT-specific sex steroid metabolism also plays a role. Cytochrome P450-dependent aromatase and 17β-hydroxysteroid dehydrogenase (17-βHSD) are two enzymes that are highly expressed in the AT stromal cells and in the preadipocytes. These enzymes are also expressed by ovarian stromal cells, and their expression is known to decrease in the menopausal state [82]. Aromatase mediates the conversion of androgens (androstenedione and testosterone) to estrogens (estrone and E_2_) (Figure 5). The enzyme 17-βHSD mediates the conversion of androstenedione (a weak androgen) and estrone (a weak estrogen) to the more potent counterparts (testosterone and E_2_, respectively). The expression of 17-βHSD is decreased relative to that of aromatase in the SAT but is greater relative to aromatase in VAT (Figure 5) [18]. The ratio of 17-βHSD to aromatase is positively associated with central adiposity due to increased local androgen production in visceral AT [83,84]. However, data on the change in relative expression of 17-βHSD and aromatase in AT in postmenopausal are currently lacking. 

Because androgens worsen insulin resistance and promote visceral adiposity, this shift in the hormone milieu from an estrogen-dominant state of premenopause to an androgen-dominant and estrogen-deplete environment of postmenopause are contributory to the metabolic transition; a worsening of insulin resistance is recognized as women transition from premenopause into postmenopause [79]. Interestingly, postmenopausal estrogen replacement therapy has been shown to mitigate this midlife metabolic transition; an attenuation in VAT accrual is observed in postmenopausal hormone users [80,85,86,87,88], supporting a role of estrogen depletion in the development of menopausal-associated metabolic dysfunction. 

## 9. Factors Influencing Adiposity during Menopausal Transition

In addition to hormonal changes, numerous other factors have been identified as influencers of adiposity during the menopausal transition. A decline in several indicators for physical performance, such as reduced handgrip strength, knee extension torque, vertical jump height, and 6-min walking distance, has been described as women transition into menopause [89]. Furthermore, multiple studies have also demonstrated that increased physical activity can mitigate this menopause transition related tendency towards increasing adiposity [90]; these findings therefore suggest that the tendency of increasing adiposity during and beyond the menopause transition may to some extent be consequent to aging related decline in physical activity. Dietary habits may also play a crucial role in influencing adiposity among postmenopausal women; indeed, inverse relationships between body adiposity and diet quality scores derived from food frequency questionnaires, encompassing the consumption of healthy foods have been demonstrated in observational studies [91,92]. Lastly, genetic underpinnings to the tendency for postmenopausal weight gain have also been suggested. While specific genetic polymorphisms linked directly to changes in adiposity during menopause in the general population remain unconfirmed, 98 independent genetic loci have been identified as influencing body fat distribution in a genome-wide association study of 362,499 individuals from the UK Biobank [93]. Among these variants, the effects of 37 associated variants were stronger in women than in men.

## 10. Conclusions

The overarching goals of this review were to help recognize AT as a complex and dynamic organ that exerts multisystem influences and that in turn is modulated and differentially regulated by the reproductive hormones. The sexual dimorphism in body fat distribution and supportive evidence of a regulatory role of estrogens, and specifically of 17β-estradiol in the metabolism and regional distribution of AT, are highlighted. Effects of aging and of menopause on AT distribution, fat mass and function are reviewed. A predilection to android fat distribution is recognized in aging women and is likely relevant to the downstream cascade of chronic health disorders including CVD and type 2 diabetes. These trends are of public health importance, making advocacy for weight management in aging women a priority area for targeted efforts aimed at harnessing this inevitable midlife spread.

## Figures and Tables

**Figure 1 ijms-25-02972-f001:**
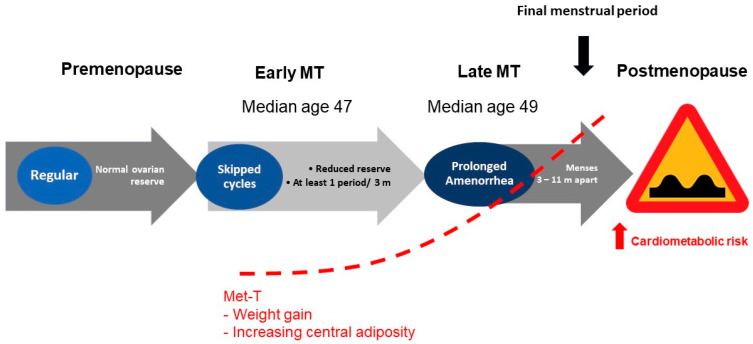
Menopause transition is accompanied by a *metabolic transition* that is characterized by a tendency towards weight gain and central adiposity. Abbreviations: MT: menopause transition; Met-T: metabolic transition; m: months.

**Figure 2 ijms-25-02972-f002:**
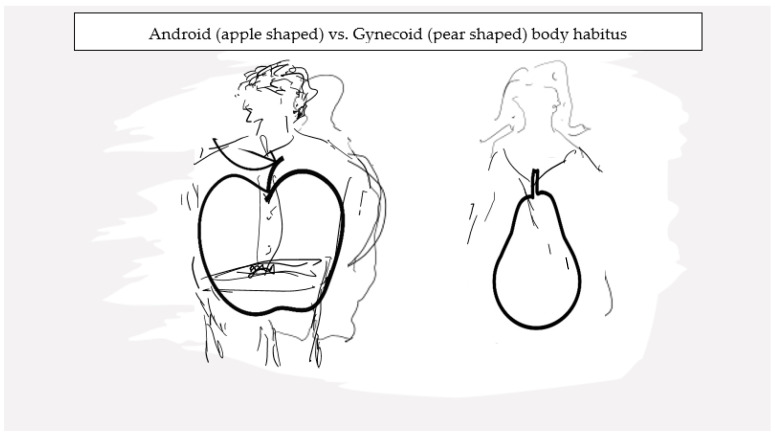
Site-specific differences in adipose tissue deposits underlie the phenotypic dimorphism between sexes.

**Figure 3 ijms-25-02972-f003:**
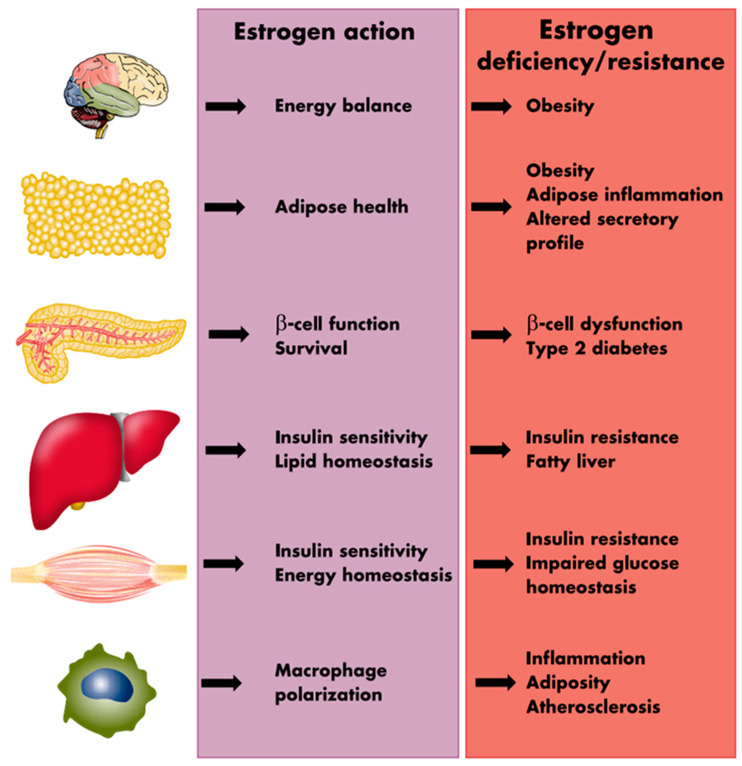
Sites of metabolic action of estrogens and metabolic sequelae of estrogen deprivation. Reprinted from Mauvais-Jarvis et al. [67]. Reprinted/adapted with permission from Oxford University Press [67]. Copyright year 2013, by the Endocrine Society.

**Figure 4 ijms-25-02972-f004:**
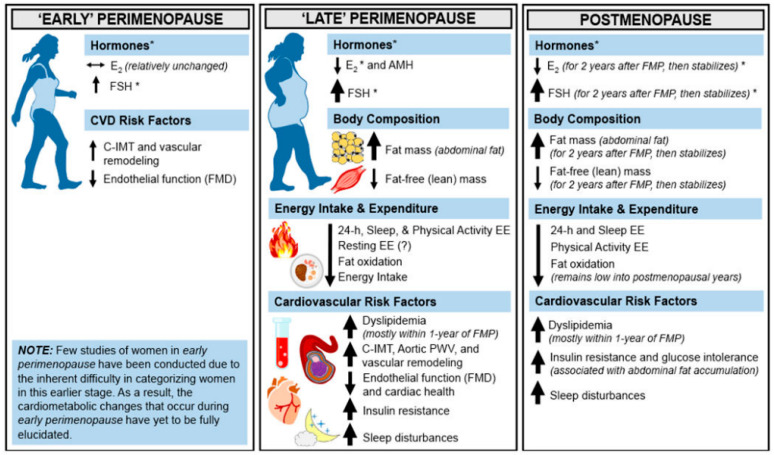
Cardiometabolic health across the menopause transition and postmenopause. The menopause transition is a distinct stage of reproductive aging and is separated into two subcategories (early perimenopause and late perimenopause). It is only after 12 months of amenorrhea following the final menstrual period that a woman is said to have arrived at menopause; then onwards, the remainder of the lifetime is spent in postmenopause. Horizontal arrows (↔) indicate stability, and smaller or larger/thicker directional arrows (↑ or ↓) indicate smaller or larger changes that occur. * Although E_2_ concentrations are lower at menopause onset compared with premenopausal concentrations, the patterns of E_2_ decline and FSH rise during perimenopause are heterogenous across women. AMH, anti-Müllerian hormone; C-IMT, carotid intima-media thickness. Abbreviations: E_2_, estradiol; EE, energy expenditure; FSH, follicle-stimulating hormone; FMP, final menstrual period; PWV, pulse wave velocity. Reprinted from Marlatt et al. [69]. Reproduced with permission from John Wiley and Sons.

**Figure 5 ijms-25-02972-f005:**
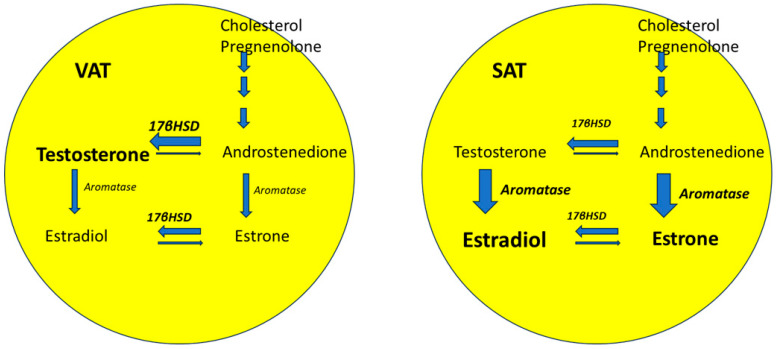
Upregulation of the enzyme (reflected by the broader size of arrow) 17β-hydroxysteroid dehydrogenase renders the visceral adipose tissue a greater source of androgen production compared to the subcutaneous adipose tissue. Abbreviations: 17βHSD: 17β-hydroxysteroid dehydrogenase; SAT: Subcutaneous adipose tissue; VAT: visceral adipose tissue.

**Table 1 ijms-25-02972-t001:** Types of Adipose Tissue.

ADIPOCYTE	WHITE	BEIGE	BROWN
**CELL TYPE**	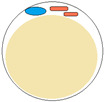	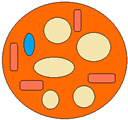	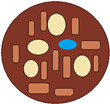
**APPEARANCE**	Unilocular lipid droplets White (WAT)	Multilocular Beige	Multilocular lipid droplets Brown (BAT)
**FUNCTION**	Energy reservoir, makes up bulk of stored adipose tissue. During high energy needs, surplus energy stored via glucose uptake and lipogenesis. Release fatty acids via lipolysis during low energy supply/increased expenditure	Bifunctional. Low basal UCP-1 Suitable for energy storage. In response to cold or sympathetic stimulation, white adipocytes increase expression of UCP-1 and release heat, “browning”	High basal UCP-1 Generates heat via uncoupling of UCP-1 in mitochondria: non shivering thermogenesis
**UCP-1 EXPRESSION**	Absent	Low	High
**MITOCHONDRIAL DENSITY**	Low	Medium	High
**LOCATION**	In the connective tissue beneath the skin (SAT) and in abdominal cavity (VAT)	WAT locations derived from white fat by “browning”	Between shoulder blades, neck, along spinal cord, and collar bone

Abbreviations: BAT: Brown adipose tissue; SAT: Subcutaneous adipose tissue; VAT: Visceral adipose tissue; WAT: White adipose tissue; UCP-1: Uncoupling protein-1.

**Table 2 ijms-25-02972-t002:** Measures for assessment of adiposity commonly used in clinical practice and in research.

Measure	Calculation	Strength	Limitation
**Body Mass Index (BMI) [20]**	- Weight (kg) divided by height (m^2^) OR - Weight (lb) divided by height (in^2^) multiplied by 703	- Simple and easy based on routinely assessed parameters - Inexpensive - Strongly correlated with BF	- Not distinguishing between BF and lean body mass - Not as accurate a predictor of BF in the elderly as it is in younger and middle-aged adults - At the same BMI, women have higher BF than men, and Asians have higher BF than Whites
**Waist Circumference (WC) [20]**	- Circumference at point between the lowest rib and the top of the hip bone) - Circumference at the narrowest point of the midsection	- Easy to measure - Inexpensive - Strongly correlated with BF in adults - Predicts development of disease and death	- Susceptible to inter-observer error based on level of measurement that must be standardized - Can be difficult to measure and less accurate in individuals of BMI ≥ 35 kg/m^2^
**Waist-to-Hip Ratio (WHR) [20]**	- Waist circumference divided by the hip circumference (at the widest diameter of the buttocks)	- Inexpensive - Good correlation with BF - Predicts development of disease and death	- Susceptible to inter-observer error based on two different measurements that must be standardized - It is easier to measure WC than the hip - Interpretation may not be easy; for example, higher WHR may represent central adiposity or alternatively may result from loss of lean gluteal muscle mass. - Measures are less accurate in individuals of BMI ≥ 35 kg/m^2^
**Skinfold Thickness (SFT) [20]**	Requires a special caliper to measure site specific thickness of a “pinch” of skin and the fat beneath Sites that can be assessed include the trunk, thighs, front and back of the upper arm, and under the shoulder blade. Equations are used to predict BF% based on SFT.	Simple Safe Portable Reusable caliper Fast and easy (except in individuals with a BMI of 35 or higher)	Lesser accuracy and reproducibility of measurements in comparison to other methods Difficult to measure in individuals of BMI ≥ 35 kg/m^2^
**Bioelectric Impedance (BIA) [20]**	- Equipment sends a small, imperceptible, safe electric current through the body, measuring the resistance. Current faces more resistance passing through BF than through lean body mass and water. - Equations are used to estimate BF % and fat-free mass.	- Safe - Relatively inexpensive - Portable - Reusable - Fast	- Expense related to purchase of equipment - Requires calibration - Ratio of body water to fat may be change during illness, dehydration or weight loss, decreasing accuracy - Lesser accuracy in comparison to other methods, especially in individuals of BMI ≥ 35 kg/m^2^
**Dual Energy X-ray Absorptiometry (DXA) [20]**	- X-ray beams pass through different tissues at different rates - DXA uses two low-level X-ray beams to develop estimates of bone mineral density, lean mass and fat mass	- Accurate - Simple - Safe	- Expense related to device and facility - Fixed, immovable - Cannot accurately distinguish between VAT and SAT - Cannot be used for pregnant women due to minimal radiation exposure - BMI limitations as most DXA equipment’s have weight/BMI limitations
**Computerized Tomography (CT) [20]**	- Highly accurate for measuring tissue, organ, and whole- BF mass, lean muscle and bone mass	- Highly accurate - Allow for measurement of fat in specific anatomical compartments, VAT, SAT	- Expense related to device and facility - Fixed, immovable - Due to radiation exposure, cannot be used in pregnant women or children - BMI limitations as most CT scanners have weight/BMI limitations
**Magnetic Resonance Imaging (MRI) [20]**	- Highly accurate for measuring tissue, organ, and whole-BF mass	- Highly accurate - Allow for measurement of fat in specific anatomical compartments, VAT, SAT - No radiation exposure/can be safely undertaken in pregnant women and children	- Expense related to device and facility - Fixed, immovable - BMI limitations as most MRI scanners have weight/BMI limitations

Abbreviations: BF: body fat; BIA: bioelectric impedance; BMI: body mass index; CT: computerized tomography; MRI: magnetic resonance imaging; SAT: subcutaneous adipose tissue; SFT: skinfold thickness; VAT: visceral adipose tissue; WC: waist circumference; WHR: waist-to-hip ratio (adapted from [20]).

**Table 3 ijms-25-02972-t003:** World Health Organization classifications of body-mass-index-based categories for non-Asian and Asian populations [28].

Categories	Non-Asians	Asians
	BMI	
**Underweight**	<18.5 kg/m^2^	<18.5 kg/m^2^
**Ideal weight**	18.5–24.9 kg/m^2^	18.5–23 kg/m^2^
**Overweight**	25–29.9 kg/m^2^	23–27.5 kg/m^2^
**Obesity**	≥30 kg/m^2^	>27.5 kg/m^2^

Abbreviations: BMI: body mass index.

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
