# Peer review of "Aging and Adiposity—Focus on Biological Females at Midlife and Beyond"

_ijms, 2024, doi:10.3390/ijms25052972_

Round 1

Reviewer 1 Report

Comments and Suggestions for Authors

The manuscript „Adiposity Across a Woman’s Life Span – relevance of chronological and reproductive aging provides valuable insights into the relationship between adiposity, aging, and menopause in women. It effectively discusses the hormonal and physiological changes during different life stages and their impact on fat distribution and metabolism. However, the manuscript could benefit from a more detailed exploration of various factors influencing adiposity, such as physical activity, dietary habits, and genetic predispositions, especially in the context of lifestyle choices and hormonal changes during menopause. Additionally, the section on energy storage and expenditure (lines 87-90) would be greatly enhanced by including specific examples and scenarios, like overnutrition or prolonged exercise, to illustrate these processes. Incorporating such examples and references to recent studies, like the one on adipokine exercise response (Int. J. Environ. Res. Public Health 2022, 19, 8782, https://doi.org/10.3390/ijerph19148782), would provide a more comprehensive understanding and make the manuscript more informative. Overall, while scientifically sound and significant, the manuscript would gain from a more integrated approach to discussing adiposity in women's health. Reviewer recommends that the authors consider these suggestions to enhance the manuscript's comprehensiveness and applicability in the field.

Author Response

Reviewer 1

The manuscript “Adiposity Across a Woman’s Life Span – relevance of chronological and reproductive aging” provides valuable insights into the relationship between adiposity, aging, and menopause in women. It effectively discusses the hormonal and physiological changes during different life stages and their impact on fat distribution and metabolism.

Response: We would like to thank the reviewer for the time and effort to review our manuscript.

However, the manuscript could benefit from a more detailed exploration of various factors influencing adiposity, such as physical activity, dietary habits, and genetic predispositions, especially in the context of lifestyle choices and hormonal changes during menopause.

Response: Thank you for the comment, we have incorporated a section discussing factors influencing adiposity during menopausal transition as suggested in the manuscript (Lines 393-410).

Additionally, the section on energy storage and expenditure (lines 87-90) would be greatly enhanced by including specific examples and scenarios, like overnutrition or prolonged exercise, to illustrate these processes. Incorporating such examples and references to recent studies, like the one on adipokine exercise response (Int. J. Environ. Res. Public Health 2022, 19, 8782, https://doi.org/10.3390/ijerph19148782), would provide a more comprehensive understanding and make the manuscript more informative.

Response: Thank you for the comment, we have included the information discussion along with the references in the manuscript (Lines 92 – 96).

Overall, while scientifically sound and significant, the manuscript would gain from a more integrated approach to discussing adiposity in women's health. Reviewer recommends that the authors consider these suggestions to enhance the manuscript's comprehensiveness and applicability in the field.

Reviewer 2 Report

Comments and Suggestions for Authors

The manuscript, submitted for evaluation titled ‘Adiposity Across a Woman’s Life Span – relevance of chronological and reproductive agingis an extensive review, collecting all relevant data on adipose tissue. The research area taken by the Authors is interesting. There are comments/questions that would be addressed to the Authors:

General Comments

Figure 1 is not fully readable.

Table 4 – it is not clear if this particular table is needed when the description is also included in the text. If it is the summary of text information why it is not in the end of the section and why it not include whole described factors?

Not necessary spaces in the text should be deleted.

The quality of figure 4 is not satisfactory. No explanation of abbreviations.

Author Response

Reviewer 2

The manuscript, submitted for evaluation titled ‘Adiposity Across a Woman’s Life Span – relevance of chronological and reproductive aging‘is an extensive review, collecting all relevant data on adipose tissue. The research area taken by the Authors is interesting. There are comments/questions that would be addressed to the Authors:

Response: We would like to thank the reviewer for the time and effort to review our manuscript.

General Comments

Figure 1 is not fully readable.

Response: We have edited Figure 1 to improved its readability.

Table 4 – it is not clear if this particular table is needed when the description is also included in the text. If it is the summary of text information why it is not in the end of the section and why it not include whole described factors?

Response: Thank you for the comment, we have now removed table 4 from the manuscript to avoid redundancy.

Not necessary spaces in the text should be deleted.

Response: We have corrected the spacing errors throughout the manuscript.

The quality of figure 4 is not satisfactory. No explanation of abbreviations.

Response: Our apologies for the incompleteness of the figure description. We have now added the description and definition of abbreviations (Lines 332-341).

Reviewer 3 Report

Comments and Suggestions for Authors

Major comments

1)    Information regarding estrogen action on LPL and HSL presented on page 10, is reiterative.

2)    All the abbreviations used in Figure 4 must be defined in the description.

3)    The review is not focused on the changes of adipose tissue biology or hormonal changes observed along a woman’s life span, it only describes some changes observed during perimenopause and postmenopausal. Furthermore, only 4 pages (of 8) are related with this topic, previous information was about general aspects that has already been largely discussed in a lot of manuscripts. In my opinion, the review is not original, not relevant and is not fully related with the proposed title.

Minor comments

1)    In table 2: add units to BMI. Also, add an “s” in “Skinfold thickness”

2)    Avoid the use of terms as “obese subjects” or “obese people”. Use instead “people with obesity”

3)    The authors use inconsistently “type II diabetes” or “type 2 diabetes”

Author Response

Reviewer 3

Major comments

1)    Information regarding estrogen action on LPL and HSL presented on page 10, is reiterative.

Response: We would like to thank the reviewer for the time and effort to review our manuscript. We have revised the discussion on estrogen action on LPL and HSL to be more concise(284 – 290).

2)    All the abbreviations used in Figure 4 must be defined in the description.

Response: Thank you for the comment, we have provided a detailed description and defined all abbreviations for Figure 4 (Lines 320 – 329).

3)    The review is not focused on the changes of adipose tissue biology or hormonal changes observed along a woman’s life span, it only describes some changes observed during perimenopause and postmenopausal. Furthermore, only 4 pages (of 8) are related with this topic, previous information was about general aspects that has already been largely discussed in a lot of manuscripts. In my opinion, the review is not original, not relevant and is not fully related with the proposed title.

Response: We appreciate the reviewer’s comments and agree that the original title of the manuscript does not fully reflect the overall content. Therefore, we have modified the title to “Aging and adiposity - focus on biological females at midlife and beyond.”

Minor comments

1)    In table 2: add units to BMI. Also, add an “s” in “Skinfold thickness”

2)    Avoid the use of terms as “obese subjects” or “obese people”. Use instead “people with obesity”

3)    The authors use inconsistently “type II diabetes” or “type 2 diabetes”

Response: We have corrected these errors in the manuscript.

Round 2

Reviewer 3 Report

Comments and Suggestions for Authors

The authors have addressed my previous comments. However, my final observation is adding units to all the BMI reported along the manuscript, i.g, in table 1. 

Also, there are some typos that should be corrected. 

Comments on the Quality of English Language

There are some typos that require correction.